# The need for speed: drivers and consequences of accelerated replication forks
Dávid Lukáč[1,2,3], Katarína Chromá [1,3] & Pavel Moudrý [1] ✉

The modulation of DNA replication dynamics has emerged as a key area of study in understanding genome stability and its perturbations in various physiological and pathological contexts. Replication fork rate is influenced by a variety of factors, including DNA repair pathways, origin firing, chromatin organization, transcription, and oncogenic signaling. This review highlights recent findings on the molecular mechanisms driving replication fork acceleration, focusing on scenarios such as PARP inhibition, oncogene activation, depletion of replication factors, and defects in Okazaki fragment processing. We discuss how reduced origin firing, R-loop resolution, and metabolic changes contribute to fork rate modulation, as well as the involvement of innate immune signaling, particularly through pathways such as cGAS-STING and ISG15. Special attention is given to consequences of accelerated replication forks for genome stability and their role in disease progression, particularly cancer. By unraveling the molecular mechanisms of fork acceleration, this Mini Review underscores its critical role in shaping genome integrity and cellular homeostasis, providing insights into future research directions and therapeutic strategies.

DNA replication, essential for cell division and the transmission of genetic information, requires precise regulation to maintain genome integrity and prevent mutations that could lead to diseases such as cancer. Replication fork (RF) progression is regulated by the coordinated action of various enzymes, including DNA polymerases, helicases, and accessory proteins. The duration of the S-phase depends on both the number of active replication origins and the rate of individual RFs, parameters that can be characterized using techniques like the DNA fiber assay. This method monitors nucleotide analog incorporation to assess RF dynamics, such as fork rate, initiation frequency, origin spacing, and fork stalling or collapse, especially under replication stress[1].

Replication stress has received significant attention in recent years, referring to conditions that impede RF progression, leading to their slowing or stalling[2,3]. Persistent stalled or collapsed forks are closely linked to genomic instability and the development of pre-cancerous and cancerous cells[4]. While replication stress traditionally involves slowing or stalling of RFs, emerging evidence suggests that even RF acceleration might also indicate ongoing replication stress[5–9]. Specifically, our study showed that poly (ADP-ribose) polymerase (PARP) inhibitors accelerate RFs and induce replication stress in human cells[5], leading to widespread interest and the identification of additional factors linked to the RF acceleration

phenomenon. In this mini review, we explore recent advancements in the field and discuss potential mechanisms driving RF acceleration.

## Origin firing as a crucial factor in replication fork dynamics

Origin firing is a crucial step in the initiation of DNA synthesis. It begins with the binding of the origin recognition complex (ORC) to specific replication origin sites, followed by the recruitment of CDC6 and CDC7 proteins. Together, the ORC and CDC6 form a complex that recruits two MCM2-7/CDT1 hexamer complexes to assemble an inactive pre-replication complex (pre-RC) at the double-stranded DNA (dsDNA). This step, known as origin licensing, occurs during the M/G1 transition and is sustained throughout the G1 phase under conditions of low cyclin-dependent kinase (CDK) activity. The transition from G1 to the S-phase activates origin firing, driven by high activity levels of DBF4-dependent kinase and CDKs. These kinases phosphorylate the MCM complex, triggering its activation. This leads to the recruitment of additional factors such as GINS and CDC45, which, along with MCM2-7, form the CDC45-MCM2-7-GINS (CMG) complex. The assembly of the CMG complex is further facilitated by key proteins, including Treslin, TopBP1, and MTBP, which are essential for recruiting GINS and CDC45. This fully formed CMG

[1]Laboratory of Genome Integrity, Institute of Molecular and Translational Medicine, Faculty of Medicine and Dentistry, Palacký University and University Hospital Olomouc, Olomouc, Czechia. [2]Department of Biochemistry and Molecular Biology, University of Southern Denmark, Odense, Denmark. [3]These authors contributed equally: Dávid Lukáč, Katarína Chromá. ✉e-mail: pavel.moudry@upol.cz

complex plays a critical role in the establishment of the pre-initiation complex (pre-IC), regulating origin firing, cell cycle progression, and activation of DNA damage checkpoints[10–12].

Reduced origin firing has been identified as a common factor contributing to accelerated RF progression across various contexts. Notably, downregulation of key pre-IC components, such as Treslin and MTBP, has been shown to increase RF rates, trigger DNA damage response (DDR), and induce asymmetric RFs[5]. Similarly, disruption of critical pre-RC components, including nascent MCMBP, leads to fork acceleration accompanied by DDR activation[13,14]. In aged hematopoietic stem cells, low levels of MCM helicase components result in faster RF progression, a higher incidence of asymmetric forks, and increased sensitivity to replication stress[6]. Notably, recent evidence indicates that an excess of loaded MCM complexes can restrain RF progression by imposing topological stress and increasing origin interference, underscoring the need for tightly regulated MCM loading to maintain proper replication dynamics[13]. Collectively, these findings highlight how insufficient availability of origin licensing factors accelerates RF progression and increases susceptibility to replication-associated stress.

## Fork rate and Okazaki fragments processing: a closer look

Okazaki fragments (OFs) are short sequences of DNA synthesized on the lagging strand during replication. Their formation requires DNA polymerase α-primase, which generates a short RNA-DNA primer to initiate synthesis[10,15]. Replication factor C (RFC) recognizes the primer, displaces the DNA polymerase α-primase complex, and recruits DNA polymerase δ. As DNA polymerase δ continues synthesizing the strand, it creates a short 5′ flap upon reaching the preceding OF. These 5′ flaps are subsequently processed by endonucleases, such as FEN1 or Dna2, which cleave the flaps, leaving nicks that are sealed by DNA ligase 1 (LIG1)[15–18].

Interestingly, depletion of FEN1 or LIG1 in human cells has been shown to increase RF rate without disrupting fork symmetry[5,19].

In *E. coli*, lagging strand synthesis can influence the rate of leading strand synthesis, making the lagging strand a limiting factor in overall RF progression[20]. While this observation originates from bacterial systems, it raises an intriguing possibility that a similar phenomenon could exist in human cancer cells. If true, FEN1- or LIG1-depleted cells may experience accelerated RF rates due to incomplete OF processing. Perhaps the maturation of OFs is a time-intensive process, potentially serving as a key determinant of RF rate. This hypothesis, however, is complicated by evidence that leading and lagging strand synthesis can be to some extent uncoupled, functioning independently of one another in certain conditions[21]. Despite this, the precise mechanism by which defective OF processing accelerates RFs remains unclear and warrants further investigation.

Interestingly, PCNA-K164R mutant 293T cells exhibit accelerated RF progression[19], phenocopying the increased fork speed observed upon depletion of key OF maturation factors such as LIG1 and FEN1[5]. These similarities extend beyond fork rate: both KR mutants and OF processing-deficient cells display nascent strand degradation under replication stress, suggesting a shared vulnerability in fork protection mechanisms. The K164 residue of PCNA is a major site for mono- and poly-ubiquitination by RAD18 and UBC13, which facilitates recruitment of translesion synthesis (TLS) polymerases and promotes post-replicative repair. Disruption of this modification compromises the recruitment of specialized polymerases, leading to the accumulation of single-stranded DNA (ssDNA) gaps that may hinder efficient OF ligation and trigger fork degradation. Supporting this, increased chromatin-associated poly-ADP-ribose chains, previously linked to OFM defects[22], are observed in both LIG1-depleted and PCNA-K164R cells[19], further reinforcing the functional overlap. Additionally, epistasis between LIG1 depletion and PCNA-K164R argues that PCNA ubiquitination and OFM operate within the same fork protection pathway[19]. These findings underscore a mechanistic link between OF processing, PCNA modification, and fork speed regulation.

## PARP inhibition and PARP-related fork rate acceleration

PARPs are multifunctional enzymes involved in key biological processes such as gene regulation, chromatin remodeling, DNA repair, and apoptosis[23]. PARP1 catalyzes the addition of ADP-ribose units to itself and other proteins through poly(ADP-ribosylation), facilitating the recruitment of repair proteins to resolve single-strand breaks, bulky lesions, double-strand breaks (DSBs), and stabilize RFs[24]. Inhibition of PARP activity impairs the repair of single-strand breaks and compromises RF stability, resulting in the accumulation of DNA damage. Consequently, PARP inhibition promotes replication stress and genomic instability, particularly in cells deficient in homologous recombination repair, such as those with BRCA1/2 mutations[24,25].

In this context, our research uncovered the unexpected finding that PARP inhibitors (PARPi), rather than stalling replication, accelerate RF progression[5]. This acceleration was accompanied by activation of the DDR, including increased γH2AX, RAD51, 53BP1, and RPA phosphorylation. We also identified p21 as a negative regulator of RF progression, with p21 depletion similarly accelerating RFs[5]. This led to the proposal of the Fork Speed Regulatory Network (FSRN), positioning PARP and the p53-p21 axis as key regulators of fork rate[5,26]. In 2021, a study from the Cantor lab revised the model of synthetic lethality in BRCA-deficient cancer cells, proposing that ssDNA gaps—rather than DSBs—are the primary drivers of PARPi cytotoxicity[7]. They showed that PARPi accelerates replication in BRCA1-deficient cells, leading to the formation of ssDNA gaps. In contrast, p21 depletion also accelerates replication but does not induce ssDNA gaps, highlighting that the presence of ssDNA gaps, rather than accelerated fork rate per se, is crucial for PARPi sensitivity[7].

It was recently shown that PARP1 provides a backup pathway for OF processing[22,27]. Since PARP1 inhibition accelerates RF progression[5] and impairs OF maturation[27], we propose that unprocessed OFs may be a key contributor to the increased fork rate observed upon PARP1 inhibition. Previous studies have demonstrated that PARPi can trap PARP1 on DNA[28–30]. This suggests that the "trapping" ability of PARPi might also immobilize OF processing enzymes, creating additional obstacles on the DNA that need to be resolved to maintain genome stability.

Further studies have linked PARPi-induced fork acceleration to the activity of PRIMPOL, a specialized polymerase involved in re-priming replication after obstacles[31]. PRIMPOL possesses both primase and polymerase activities, allowing it to initiate new primers downstream of DNA lesions or structures that impede fork progression. This re-priming activity enables fork continuation at the cost of leaving behind post-replicative ssDNA gaps, a mechanism that is well established[32–40]. PRIMPOL depletion prevents PARPi-induced fork acceleration[41,42], suggesting that PARP inhibition triggers PRIMPOL-mediated re-priming, suppresses fork reversal, and promotes ssDNA gap formation in response to DNA damage. While the mechanistic link between PRIMPOL and ssDNA gaps is clear, how PRIMPOL activity leads to RF acceleration remains poorly understood.

Our recent work highlights the DNA polymerase α/primase in PARPi-induced fork acceleration, showing that it promotes fork acceleration; however, in contrast to PRIMPOL, DNA polymerase α/primase prevents ssDNA gap formation[42]. Additionally, Giansanti and colleagues found that MDM2 overexpression accelerates RF progression via PARP1 destabilization. Strikingly, both PARPi- and MDM2-induced RF accelerations are dependent on RECQ1 and PRIMPOL[41]. MDM2 is an E3 ubiquitin ligase best known for its role in negatively regulating the tumor suppressor p53 by targeting it for proteasomal degradation[43]. In the context of DNA replication, MDM2 influences fork dynamics through two distinct mechanisms. First, it promotes PARP1 destabilization, contributing to fork acceleration[41]. Second, MDM2 suppresses p53 levels, which also impacts replication speed[5,44]. Castaño and colleagues showed that reduced p53 levels lead to faster fork progression and increased replication stress, and that p53 levels modulate the usage of DNA damage tolerance pathways[44]. These findings highlight the dual role of MDM2 in modulating replication dynamics and reinforce the importance of both PARP1 and p53 in the FSRN.

CARM1 deficiency also accelerates RFs[45]. CARM1 (coactivator-associated arginine methyltransferase 1) is a chromatin-associated regulator that exerts a methyltransferase-independent function at DNA RFs to control fork speed and stress response pathway choice[45]. CARM1 interacts with PARP1 to promote its activation, which in turn inhibits the helicase RECQ1, a key driver of RF restart. RECQ1 belongs to the RecQ family of DNA helicases and plays a central role in restoring reversed RFs by catalyzing branch migration. By promoting fork restart, RECQ1 helps resume DNA synthesis and prevents prolonged fork stalling, thereby maintaining replication efficiency and genome stability[45]. CARM1-mediated inhibition of RECQ1 by PARP1 favors fork reversal and limits the engagement of PRIMPOL and TLS, thereby slowing RF progression and preserving replication fidelity. In the absence of CARM1, PARP1 activation is impaired, leading to deregulated RECQ1 activity, premature fork restart, and increased reliance on PRIMPOL and TLS, resulting in accelerated but potentially less accurate DNA replication[45].

Similarly, ten-eleven translocation 2 (TET2) is a dioxygenase that catalyzes the oxidation of 5-methylcytosine, initiating active DNA demethylation and playing a key role in epigenetic regulation[46]. TET2 deletion has been shown to accelerate RFs and impair DSB repair, leading to increased genomic instability[47]. These defects result in a synthetic lethality phenotype when TET2-deficient cells are treated with PARPi[47], highlighting a critical role for TET2 in maintaining genome integrity during replication stress.

## cGAS–STING pathway's impact on fork rate acceleration

The cGAS–STING pathway is integral to innate immunity, sensing both foreign and self-DNA to mediate various cellular responses ranging from infection defense to inflammatory disease pathogenesis. When dsDNA enters the cytoplasm, cyclic GMP-AMP synthase (cGAS) binds to it, forming dimers and activating its enzymatic function to produce cyclic GMP-AMP (cGAMP). cGAMP binds to stimulator of interferon genes (STING) at the endoplasmic reticulum, inducing its conformational changes. Activated STING recruits TANK-binding kinase 1 (TBK1), which phosphorylates both STING and interferon regulatory factor 3 (IRF3). Phosphorylated IRF3 translocates to the nucleus, where it induces expression of immune response genes, including type I interferons, interferon-stimulated genes (ISGs), pro-apoptotic factors, chemokines, and other inflammatory mediators[48].

In addition to its immune role, cGAS also functions in the nucleus independently of cGAMP and STING. A recent study revealed that chromatin-bound cGAS can directly interact with core replication factors, including MCM2, MCM7, RFC1, and PCNA[49]. Through this non-canonical role, cGAS acts as a physical "roadblock" on DNA to restrain RF speed and limit unscheduled origin firing. Accelerated RF progression combined with deregulated origin firing in cGAS-deficient cells ultimately fuels genomic instability[49].

ISG15, one of the most strongly induced ISGs, is a ubiquitin-like protein modifier that can be conjugated to target proteins via a process known as ISGylation. Beyond its canonical antiviral functions, ISG15 has been identified as a critical regulator of RF dynamics. High levels of ISG15, either intrinsic or induced by interferon-β, accelerate RF progression without affecting origin firing or fork symmetry and result in DNA damage and chromosomal aberrations[8]. This acceleration is mediated through ISG15's interaction with RECQ1[8], a DNA helicase responsible for restarting stalled forks, enhancing its activity to facilitate faster fork movement. While the precise mechanism remains unclear, involvement of RECQ1 suggests a shared pathway with PARPi, both of which appear to modulate fork dynamics by deregulating RECQ1 activity[8,41]. Interestingly, although cGAS deficiency and ISG15 overexpression are opposite perturbations in the cGAS–STING–ISG15 axis, both result in accelerated fork progression. This apparent contradiction can be resolved by considering that cGAS restrains replication independently of its immune signaling function, whereas ISG15 acts downstream of interferon signaling to modulate fork dynamics via protein interactions. In cGAS-deficient cells, the absence of chromatin-bound cGAS removes physical constraints on RFs. In contrast, ISG15-driven acceleration reflects a distinct, interferon-dependent pathway that is dependent on RECQ1 activity[8]. Notably, ISG15 expression is elevated in BRCA-deficient cells, which exhibit rapid PARPi-induced fork acceleration[7,50]. This raises the possibility that PARPi promote RF acceleration via ISG15. Indeed, PARPi triggers the accumulation of cytosolic DNA and activates the cGAS–STING pathway[51,52]. Whether PARPi-induced fork acceleration is directly mediated by ISG15 or involves a broader innate immune response remains an open question requiring further investigation.

## Oncogene signaling as a driver of replication fork acceleration

Oncogenes are genes that, when mutated or abnormally activated, drive cancer development by promoting unchecked cell proliferation, inhibiting apoptosis, or disrupting normal cellular functions[53]. These genes play a pivotal role in cancer initiation and progression. While oncogene activation is not a direct mechanism of RF acceleration, it can trigger transcriptional, metabolic, and replication-related changes that engage molecular pathways promoting faster fork progression. A key consequence of oncogene signaling is replication stress, which often arises through altered origin firing, transcription–replication conflicts, or metabolic reprogramming[54].

The RAS oncogene encodes a small GTPase that transduces signals from growth factor receptors to downstream effectors such as the MAPK and PI3K/AKT pathways, regulating cell proliferation and metabolism[55].

**Table 1 | Proteins whose manipulation leads to the acceleration of replication forks**

| Proteins | Manipulation leading to fork acceleration | Cell lines | References |
|---|---|---|---|
| PARP1 | Depletion, Inhibition | U2OS, HeLa, MDA-MB-436, OVCAR-5, BJ | 5 |
| FEN1, LIG1 | Depletion | U2OS, RPE1, 293T | 5,19,70 |
| p21 | Depletion | U2OS, RPE1 | 5,7 |
| MDM2 | Induced accumulation[a] | SJSA-1, H1299, RPE1 | 41 |
| cGAS | Depletion | BJ | 49 |
| ISG15 | Overexpression[b] | U2OS | 8 |
| CARM1 | Depletion | MCF10A, MEF, U2OS | 45 |
| TET2 | Depletion | Lin-cKitt[+] BMC | 47 |
| RAS | Overexpression | BJ, FSE-hTert, WI38-hTert | 9,56 |
| RNAP2 | Inhibition[c] | RPE1 | 66 |
| PCID2, GANP | Depletion | HeLa | 69 |
| THOC1 | Depletion | HeLa | 67 |
| Treslin, MTBP | Depletion | U2OS | 5 |
| MCM4 | Depletion | U2OS | 13 |
| Cyclin D | Overexpression, AMBRA1 depletion | BJ-hTERT | 58 |
| E2F | E2F6 depletion | RPE1 | 59 |
| SPI1 | Overexpression | K562 | 57 |
| SHh | Stimulation | GCP | 61 |

The table summarizes key studies reporting proteins whose altered expression or activity results in increased replication fork speed. The table includes the specific protein involved, the type of manipulation performed, the cellular model used, and the corresponding references.
Special considerations: [a]Includes both MDM2 overexpression and Nutlin-induced MDM2 accumulation. [b]Acceleration is independent of ISG15 conjugation. [c]Effect is restricted to the early S-phase.

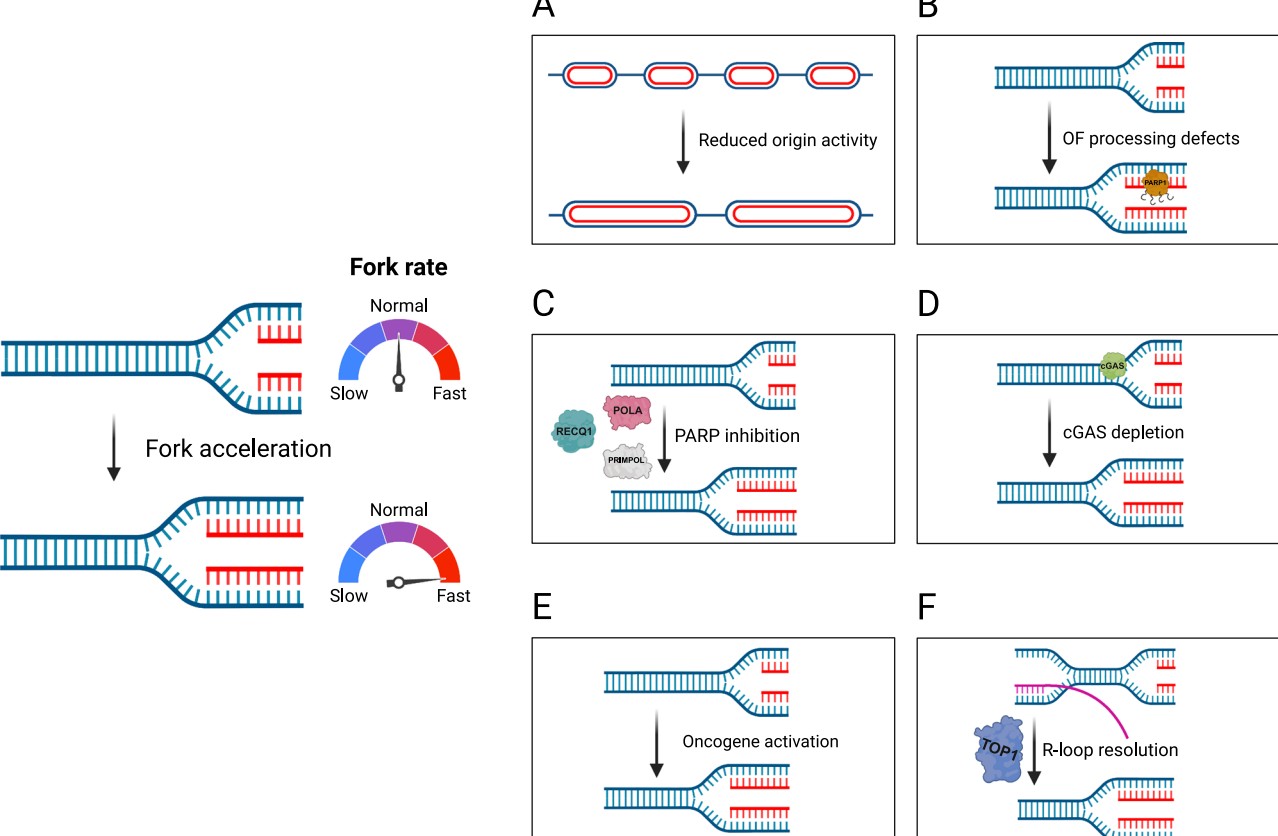

**Fig. 1 | Mechanisms and factors driving replication fork acceleration.** Replication fork acceleration beyond the normal physiological range can result from diverse perturbations. **A**–**F** Key pathways and molecular alterations associated with increased fork speed. **A** Reduced origin activity leads to increased inter-origin distance and compensatory acceleration of fork progression. **B** Disruption of Okazaki fragment processing, including depletion of LIG1 or FEN1, or mutation of PCNA at K164, leads to fork acceleration and increased replication stress. **C** PARP inhibition promotes fork acceleration through mechanisms involving PRIMPOL-, POLA-, and RECQ1-dependent pathways. **D** cGAS depletion removes a replication-associated barrier, increasing fork rate by disrupting cGAS interactions with replication proteins. **E** Activation of oncogenes, such as by RAS, SPI1, or SHH, modulates transcriptional and metabolic programs that influence replication dynamics and accelerate replication forks. **F** TOP1-mediated R-loop resolution limits replication-transcription conflicts, thereby increasing replication fork speed.

Maya-Mendoza and colleagues demonstrated that RAS signaling initially induces RF acceleration, followed by a reduction in RF rate and oncogene-induced senescence due to progressive metabolic exhaustion[56].

The transcription factor SPI1, a master regulator of hematopoiesis, can also act as an oncogene. Its overexpression promotes RF acceleration[57]. Notably, SPI1-induced acceleration occurs without a decrease in origin activity, accumulation of asymmetric forks, or activation of the DDR, suggesting a transcription-mediated mechanism that bypasses conventional replication stress pathways.

Similarly, upregulation of Cyclin D activity, which promotes G1-S transition and is frequently overexpressed in cancers, accelerates RF progression either through AMBRA1 depletion or Cyclin D overexpression[58]. In addition, elevated E2F activity, driven by depletion of the transcriptional repressor E2F6, accelerates RFs[59]. As E2F transcription factors are key regulators of S-phase entry and frequently deregulated in cancer, their influence on replication dynamics underscores their oncogenic potential.

The Sonic Hedgehog (SHh) signaling pathway, essential for embryonic development and tissue patterning, is aberrantly activated in several cancers. SHh signaling operates through the Smoothened receptor and downstream GLI transcription factors, driving gene expression that supports proliferation and survival[60]. In granule cell progenitors (GCPs), Tamayo-Orrego and colleagues showed that SHh stimulation increases RF rates without changes in fork symmetry or stalling[61]. Intriguingly, SHh stimulation not only accelerated RFs but also promoted origin firing. The enhanced fork progression in SHh-stimulated cells is attributed to increased expression of genes involved in nucleotide metabolism, pre-RC formation, licensing, and

helicase activity. Meanwhile, the rise in origin firing is directly driven by SHh-induced activation of CDC7, a key regulator of origin firing[61]. These changes collectively shortened S-phase duration, suggesting that SHh-stimulated cells optimize replication efficiency by simultaneously accelerating RFs and increasing origin firing. This is particularly noteworthy, as RF rate and origin firing are typically inversely correlated[62]. The authors speculate that the dual increase is likely an adaptive mechanism for the rapid expansion of GCPs, the most abundant neural cell type in the human brain. Whether this strategy is unique to GCPs or applies more broadly remains an open question.

## Transcription-associated mechanisms: R-loops and fork dynamics

R-loops are three-stranded nucleic acid structures that form during transcription when the nascent RNA hybridizes back to the DNA template, displacing the non-template strand. While R-loops play physiological roles in gene regulation and genome stability, their unscheduled accumulation poses a threat to DNA replication by creating physical barriers to RF progression. Excessive R-loop formation can stall forks, promote DNA breaks, and trigger replication stress[63]. Cells deploy multiple mechanisms to regulate R-loop levels, including RNase H enzymes, helicases, and topoisomerases. In particular, topoisomerase 1 (TOP1) resolves transcription-induced supercoiling, preventing the excessive torsional stress that favors R-loop formation[64]. Thus, TOP1 activity is central to maintaining the balance between transcription and replication by mitigating R-loop-associated fork obstacles.

## Box 1 | Accelerated replication forks—key advances, open questions, and technical challenges

*Key advances*
- Fork acceleration has emerged as a distinct replication phenotype, characterized by replication forks progressing faster than under normal, unchallenged conditions.
- Reduced origin activity, oncogene activation, cGAS deficiency, ISG15 overexpression, and defects in Okazaki fragment maturation have all been associated with increased replication fork rates.
- Specialized factors such as PRIMPOL, RECQ1, and DNA polymerase α-primase, implicated in fork restart and gap tolerance, contribute to fork acceleration, including in response to PARP inhibitors.

*Outstanding questions*
- What upstream signals stimulate PRIMPOL, RECQ1, and DNA polymerase α-primase, and what are the precise molecular mechanisms by which they promote fork acceleration across different cellular contexts?
- What are the molecular mechanisms underlying how low origin firing activity leads to replication fork acceleration, and conversely, how accelerated forks influence the regulation of origin activity?

- How do distinct acceleration mechanisms converge or differ in their impact on genome stability?

*Technical challenges*
- While DNA fiber assays and single-molecule approaches have the potential to distinguish continuous from discontinuous fork progression, many published studies lack experimental designs that clearly address this distinction.
- Quantifying replication speed in heterogeneous cell populations remains difficult; current methods rely exclusively on manual image acquisition and analysis, making them time-consuming and susceptible to user bias.
- Disentangling fork acceleration from changes in origin usage, fork protection, or DNA repair requires combined structural, biochemical, and live-cell imaging approaches.

Consistent with the findings of Maya-Mendoza and colleagues[56], RAS-expressing cells exhibit accelerated RF rates associated with an increased inter-origin distance while maintaining fork symmetry[9]. Although reduced origin firing is known to accelerate RFs[6,13], no significant changes were observed in the expression of genes associated with origin firing or licensing[9]. Interestingly, RAS-expressing cells display decreased levels of R-loops—structures known to impede fork progression and slow replication. This reduction in R-loops has been proposed as a key factor driving the increased fork rate. Supporting this, RAS-expressing cells show elevated levels of TOP1, and its overexpression accelerates RF progression and increases inter-origin distance[9]. These findings align with studies showing that loss of TOP1 leads to R-loop accumulation in actively transcribed genes, resulting in fork stalling and slowed replication[65].

The connection between R-loops and RF rate underscores the key role of transcription in regulating DNA replication dynamics. Extensive transcriptional activity, especially in early S-phase, limits RF progression[66]. RF rate increases when transcription is inhibited in early S-phase, possibly due to reduced transcription–replication interference[66]. Further supporting this, depletion of the THOC1, a subunit of the human transcription and ribonucleoprotein export complex THO/TREX, accelerates RF progression[67]. This may reflect reduced RNAP2 activity, longer replicons, or chromatin hyperacetylation[67,68]. Similarly, depletion of PCID2 and GANP, subunits of the TREX2 complex, also accelerates RFs[69]. The authors proposed that in the absence of TREX2, other RNA-binding proteins prevent R-loop accumulation, further implicating R-loops as negative regulators of RF progression. This idea is supported by the observation that preventing R-loop accumulation through multiple interventions results in increased RF rate[9,69].

### Outlook and conclusions

The recent findings outlined above illuminate the complex network of factors that regulate RF dynamics. A diverse array of proteins involved in processes such as DNA repair, cell cycle regulation, and innate immune signaling converge to fine-tune the rate of DNA replication, as summarized in Table 1 and Fig. 1. The intricate interplay between replication dynamics and cellular metabolism, oncogenic signaling, and transcriptional activity adds another layer of complexity to our understanding of DNA replication dynamics.

Future investigations will likely focus on uncovering the molecular mechanisms that bridge these diverse processes with RF regulation (Box 1). For instance, exploring how specific DNA repair pathways, such as homologous recombination and nucleotide excision repair, intersect with replication dynamics may help to understand the coordination between replication and repair systems. Similarly, studying the metabolic alterations induced by oncogenic signaling pathways and their impact on DNA replication may reveal novel therapeutic vulnerabilities. Additionally, characterizing the role of chromatin modifiers and epigenetic regulators in RF dynamics may open new avenues for targeting DNA replication processes in cancer therapy. In conclusion, the studies reviewed here emphasize the key role of RF dynamics in maintaining genome stability and cellular homeostasis. Deciphering the regulatory networks of DNA replication provides a deeper understanding of the molecular underpinnings of diseases like cancer and could lead to innovative strategies for intervention and treatment.

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

## Acknowledgements
This article was supported by the SALVAGE project (registration number: CZ.02.01.01/00/22_008/0004644, supported by OP JAK, with co-financing from the EU and the State Budget), the MEYS CR through large RI projects EATRIS (EATRIS-CZ LM2023053) and Czech-BioImaging (LM2023050), and an internal grant of Palacký University (IGA_LF_2025_025).

## Author contributions
D.L.: writing—original draft, writing—reviewing and editing. K.C.: writing—original draft. P.M.: conceptualization, writing—original draft, writing—reviewing and editing, supervision.

## Competing interests
The authors declare no competing interests.
