## [Transparent Peer Review file · Communications Biology]

The Need for Speed: Drivers and Consequences of Accelerated Replication Forks

Corresponding Author: Dr Pavel Moudry

Version 0:

Reviewer comments:

Reviewer #1

(Remarks to the Author)

This is a review about replication fork acceleration as a mechanism of replication stress. It is timely and relevant to publish a review on this topic and the work could be of broad interest. The supplied table could provide an excellent resource and overview.

However, the review covers a complex subject and in its current form would need extensive re-structuring, a change emphasis and much additional content to do this justice.

Figure 1

- Is much too basic and should provide a schematic of key mechanisms.

Table 1

- Should clearly indicate that the Manipulation in the central column leads to fork acceleration in all cases. It would be useful to add a column listing the cell line/model used in each study, and a column adding any special considerations (e.g. RNAP2 inhibition in early S phase).

Overall structure

- Fork acceleration has been linked to the regulation and activity of specific enzymes, such as PRIMPOL and RECQ1. There should be additional sections providing background on these enzymes and their functions at replication forks. For example, the link between PRIMPOL and ssDNA gaps needs to be explained.

- Similarly, it should be explained what key factors such as CARM1, TET2 etc are.

- It would be more logical to start the main body text with 1) origin firing and 2) Okazaki fragment maturation, followed by perhaps sections on PRIMPOL and RECQ1 etc, then PARP, cGAS and oncogenes.

- Mechanistic detail as to how e.g. PARP activity might restrain fork speeds that are e.g. in the Okazaki section should be moved into the PARP section.

PARP

- The content on PARP is too focused on the author's own work. For example, in the Introduction it is stated that: "emerging evidence suggests that even replication fork acceleration might also indicate ongoing replication stress (5)." Here only the author's own work is cited, but there are additional studies linking fork acceleration to replication stress, which should be cited here to support the assertion that evidence is emerging.

- It should be briefly explained what the major cellular effects of PARP inhibition are (e.g. inhibited DNA repair, increased DNA damage, etc).

P53

- Castano NAR 2024 (PMID 38321962) reporting that p53 levels can modulate fork speed should be cited.

- The link between MDM2 and p53 should be made clear.

Oncogenes

- Oncogene signaling is not a mechanism of fork acceleration in its own right. Rather, oncogene signaling can activate mechanisms that accelerate forks. It would be preferable to clearly distinguish between actual mechanisms of fork acceleration and cellular signaling pathways that are shown to influence these mechanisms, such as oncogene signaling and cGAS/ISG15.

- The main signalling functions and downstream effects of RAS, SPI1, Shh etc should be mentioned.

Transcription

- A separate section on transcription and R-loops should be considered as not all oncogene effects are linked to transcription.
- R-loops and the role of TOP1 in their control should receive a proper background introduction, like origin firing and Okazaki fragment processing in sections 4 and 5.
- The paragraph lines 155-167 is completely unclear without a proper introduction of the relationship between transcription and R-loops, and of R-loop regulation.

cGAS

- It is not clear how the pathways cGAS activation relates to the mechanism whereby cGAS directly interacts with the replication machinery. Does cGAS need to be activated to act as a “roadblock”?
- The function of ISG15 as a small protein modifier should be introduced.
- It is curious that cGAS deficiency and high levels of ISG15 both lead to fork acceleration and these are diametrically opposed interventions in the cGAS-ISG15 pathway. This should be discussed.

Reviewer #2

(Remarks to the Author)

In this short manuscript, Lukac et al. review different mechanisms of fork acceleration reported so far, focusing on PARP inhibition and encompassing the cGAS-STING pathway, oncogene activation, transcription, origin usage and Okazaki fragment processing. The regulation of fork speed is an emerging concept for which Dr. Moudry contributed a seminal paper in Nature (Maya-Mendoza et al., 2018), reporting for the first time the existence of a network controlling fork speed that involves PARP1, p21 and p53.

This review is very well written and mostly comprehensive. The authors not only summarize the current state of knowledge on the control over fork speed but also give their perspective on the pressing issues that need to be addressed to gain a deeper understanding of the mechanisms involved and thus design better cancer therapies.

Major comment:

1. The Fork Protection Complex (FPC), especially the CLASPIN and TIMELESS proteins, has known roles in the control of replisome progression, i.e. fork speed (Yeeles et al., Mol Cell 2017 [PMID: 27989442], Baris et al., Nature 2022 [PMID: 35585232], Somyajit et al., Science 2017 [PMID: 29123070], etc..). It probably would deserve a separate section as it represents another mechanism of control over fork speed.

Minor comment:

1. Regarding the section on origin firing, it has been shown that the excess of MCM proteins loaded on origins actually restrains replication fork progression (Sedlackova et al., Nature 2020, PMID: 33087936). Can the authors discuss this reference at the end of the section, as it reinforces the point made with aged HSCs?

2. Please cite the Maya-Mendoza et al., 2018 study when you refer to your study line 41.

Reviewer #3

(Remarks to the Author)

This review article discusses recent advances in understanding replication stress and the mechanisms that regulate fork progression. It emphasizes the impact of PARP1 inhibition, the cGAS-STING pathway, oncogenes, transcription-replication conflicts, and origin firing on fork dynamics.

This review article provides valuable insights into the current understanding of the mechanisms underlying replication stress and provides potential directions for future research. In the section on PARP1 inhibition, several studies have demonstrated that PARP1/DNA translocases such as HLTF, SMARCAL1, and ZRANB3, facilitate fork reversal, which can influence fork speed. The roles of RAD18, UBC13, PCNA ubiquitination, and TLS polymerases in gap filling can also be included in this review.

Version 1:

Reviewer comments:

Reviewer #1

(Remarks to the Author)

The authors have addressed my comments well and the review is now much better structured. As a final improvement to the section on oncogene signaling and Table 1, the authors should also consider including reported effects of up-regulating Cyclin D activity (through AMBRA1 depletion or Cyclin D1 overexpression, PMID: 33854232) and E2F activity (through E2F6 depletion, PMID: 32665547) on fork acceleration.

Reviewer #2

(Remarks to the Author)

The authors have answered my concerns and greatly improved the flow and the clarity of the manuscript. Thank you!

Response to Reviewers' comments

We would like to thank you and the reviewers for your thorough evaluation of our manuscript and for the valuable, constructive feedback. We have revised the manuscript accordingly and provide detailed responses below to each of the comments raised. These revisions have helped us improve the clarity, structure, and focus of the Mini Review.

Please note that this Mini Review is specifically centered on mechanisms that accelerate replication fork progression under unchallenged conditions, rather than those involved in general fork stability or restraint. As such, some topics—while highly relevant to fork dynamics—were not included in dedicated sections due to both scope and space limitations. These decisions are justified below, and we remain fully open to incorporating additional discussion if the editor or reviewers consider it necessary.

Reviewer #1 (Remarks to the Author):

This is a review about replication fork acceleration as a mechanism of replication stress. It is timely and relevant to publish a review on this topic and the work could be of broad interest. The supplied table could provide an excellent resource and overview.

However, the review covers a complex subject and in its current form would need extensive re-structuring, a change emphasis and much additional content to do this justice.

Response: Thank you for this feedback. We have revised the manuscript extensively to improve its organization and clarity, and we have added several explanatory details throughout. We also adjusted the structure to improve the logical flow of the narrative, while remaining within the constraints of the Mini Review format.

Figure 1

- Is much too basic and should provide a schematic of key mechanisms.

Response: We have revised Figure 1 to include a schematic overview of the key mechanisms described in the review, in line with the suggestion.

Table 1

- Should clearly indicate that the Manipulation in the central column leads to fork acceleration in all cases. It would be useful to add a column listing the cell line/model used in each study, and a column adding any special considerations (e.g. RNAP2 inhibition in early S phase).

Response: We have added a column specifying the cell line or model system used in each cited study. For special considerations (e.g., timing of treatment), we used superscript letters

(a, b, c) linked to notes in the figure legend, rather than a separate column, to preserve formatting and readability.

Overall structure

- Fork acceleration has been linked to the regulation and activity of specific enzymes, such as PRIMPOL and RECQ1. There should be additional sections providing background on these enzymes and their functions at replication forks. For example, the link between PRIMPOL and ssDNA gaps needs to be explained.

Response: Background information on PRIMPOL and RECQ1 has been added to relevant parts of the manuscript. However, current evidence indicates that activation/overexpression of these factors does not lead to fork acceleration. For example, studies by Quinet et al. 2020 (PMID: 31676232) and Mehta et al. 2022 (PMID: 35353580) found no significant change in fork speed upon PRIMPOL overexpression. Therefore, we have chosen not to include them as standalone sections, both due to the limited supporting data and the format constraints of the Mini Review.

- Similarly, it should be explained what key factors such as CARM1, TET2 etc are.

Response: Thank you - brief explanations of CARM1 and TET2 have been added to the revised manuscript.

- It would be more logical to start the main body text with 1) origin firing and 2) Okazaki fragment maturation, followed by perhaps sections on PRIMPOL and RECQ1 etc, then PARP, cGAS and oncogenes.

Response: We have revised the structure of the manuscript to follow the suggested logical sequence. As noted, PRIMPOL and RECQ1 were not included as separate sections, but relevant information has been incorporated where appropriate.

- Mechanistic detail as to how e.g. PARP activity might restrain fork speeds that are e.g. in the Okazaki section should be moved into the PARP section.

Response: We have relocated the mechanistic discussion of PARP activity to the appropriate section.

PARP

- The content on PARP is too focused on the author's own work. For example, in the Introduction it is stated that: "emerging evidence suggests that even replication fork

acceleration might also indicate ongoing replication stress (5).” Here only the author’s own work is cited, but there are additional studies linking fork acceleration to replication stress, which should be cited here to support the assertion that evidence is emerging.

Response: We agree and have added references to additional studies that support the emerging link between replication fork acceleration and replication stress.

- It should be briefly explained what the major cellular effects of PARP inhibition are (e.g. inhibited DNA repair, increased DNA damage, etc).

Response: A brief summary of the main cellular consequences of PARP inhibition has been added for context.

P53

- Castano NAR 2024 (PMID 38321962) reporting that p53 levels can modulate fork speed should be cited.

- The link between MDM2 and p53 should be made clear.

Response: We have cited the suggested reference and clarified the relationship between MDM2 and p53 in the revised section.

Oncogenes

- Oncogene signaling is not a mechanism of fork acceleration in its own right. Rather, oncogene signaling can activate mechanisms that accelerate forks. It would be preferable to clearly distinguish between actual mechanisms of fork acceleration and cellular signaling pathways that are shown to influence these mechanisms, such as oncogene signaling and cGAS/ISG15.

Response: We have revised this section to clarify that oncogene signaling acts upstream of the mechanisms that directly drive fork acceleration. Descriptions of the main functions and downstream effects of RAS, SPI1, and Shh have also been added.

- The main signalling functions and downstream effects of RAS, SPI1, Shh etc should be mentioned.

Response: Descriptions of the main functions and downstream effects of RAS, SPI1, and Shh have also been added.

Transcription

- A separate section on transcription and R-loops should be considered as not all oncogene effects are linked to transcription.

- R-loops and the role of TOP1 in their control should receive a proper background introduction, like origin firing and Okazaki fragment processing in sections 4 and 5.

Response: We have created a separate paragraph on transcription and R-loops and expanded the background information to contextualize their relationship to replication stress.

- The paragraph lines 155-167 is completely unclear without a proper introduction of the relationship between transcription and R-loops, and of R-loop regulation.

Response: This paragraph has been revised, and additional background on R-loop regulation has been introduced to improve clarity.

cGAS

- It is not clear how the pathways cGAS activation relates to the mechanism whereby cGAS directly interacts with the replication machinery. Does cGAS need to be activated to act as a "roadblock"?

Response: We clarified that cGAS's role at replication forks appears to be independent of canonical activation and more dependent on chromatin localization and direct interactions.

- The function of ISG15 as a small protein modifier should be introduced.

- It is curious that cGAS deficiency and high levels of ISG15 both lead to fork acceleration and these are diametrically opposed interventions in the cGAS-ISG15 pathway. This should be discussed.

Response: ISG15 is now introduced as a ubiquitin-like modifier, and we have addressed the apparent paradox of cGAS deficiency and ISG15 overexpression both leading to fork acceleration.

Reviewer #2 (Remarks to the Author):

In this short manuscript, Lukac et al. review different mechanisms of fork acceleration reported so far, focusing on PARP inhibition and encompassing the cGAS-STING pathway,

oncogene activation, transcription, origin usage and Okazaki fragment processing. The regulation of fork speed is an emerging concept for which Dr. Moudry contributed a seminal paper in Nature (Maya-Mendoza et al., 2018), reporting for the first time the existence of a network controlling fork speed that involves PARP1, p21 and p53.

This review is very well written and mostly comprehensive. The authors not only summarize the current state of knowledge on the control over fork speed but also give their perspective on the pressing issues that need to be addressed to gain a deeper understanding of the mechanisms involved and thus design better cancer therapies.

Major comment:

1. The Fork Protection Complex (FPC), especially the CLASPIN and TIMELESS proteins, has known roles in the control of replisome progression, i.e. fork speed (Yeeles et al., Mol Cell 2017 [PMID: 27989442], Baris et al., Nature 2022 [PMID: 35585232], Somyajit et al., Science 2017 [PMID: 29123070], etc..). It probably would deserve a separate section as it represents another mechanism of control over fork speed.

Response: We appreciate the reviewer's suggestion and agree that the FPC plays an essential role in maintaining replication fork stability. However, our review specifically focuses on factors that increase fork speed beyond physiological baseline levels in unchallenged cells. To our knowledge, FPC components have not been shown to cause acceleration of fork speed when overexpressed or activated.

Moreover, due to the concise format of the Mini Review, we prioritized mechanisms with direct, experimentally demonstrated effects on fork acceleration. For these reasons, we did not include a dedicated section on the FPC. However, we would be happy to include a brief discussion if the reviewer or editor feels it would enhance the manuscript's balance.

Minor comment:

1. Regarding the section on origin firing, it has been shown that the excess of MCM proteins loaded on origins actually restrains replication fork progression (Sedlackova et al., Nature 2020, PMID: 33087936). Can the authors discuss this reference at the end of the section, as it reinforces the point made with aged HSCs?

Response: Thank you for this valuable suggestion. We have cited the Sedlackova et al. study and briefly discussed the finding in the context of fork restraint at the end of the origin firing section.

2. Please cite the Maya-Mendoza et al., 2018 study when you refer to your study line 41.

Response: The Maya-Mendoza et al. (2018) study has now been cited in the relevant section.

Reviewer #3 (Remarks to the Author):

This review article discusses recent advances in understanding replication stress and the mechanisms that regulate fork progression. It emphasizes the impact of PARP1 inhibition, the cGAS-STING pathway, oncogenes, transcription-replication conflicts, and origin firing on fork dynamics.

This review article provides valuable insights into the current understanding of the mechanisms underlying replication stress and provides potential directions for future research. In the section on PARP1 inhibition, several studies have demonstrated that PARP1/DNA translocases such as HLF1, SMARCA1, and ZRANB3, facilitate fork reversal, which can influence fork speed.

Response: We thank the reviewer for highlighting this important set of proteins. These factors are indeed critical for replication fork reversal and maintaining fork stability under stress. However, there is no direct evidence that their depletion or loss results in fork acceleration under normal conditions. Our own unpublished data support this, showing no fork acceleration upon downregulation of fork reversal factors in unchallenged cells. Due to the scope and space constraints of the Mini Review, we did not include a separate discussion on fork reversal. However, we would be happy to include a brief discussion in the text if preferred.

The roles of RAD18, UBC13, PCNA ubiquitination, and TLS polymerases in gap filling can also be included in this review.

Response: These components are now briefly discussed in Chapter 2 of the revised manuscript, as they are relevant to the regulation of fork progression through gap filling and post-replicative repair pathways.

We sincerely thank all reviewers for their constructive and thoughtful feedback, which has helped us improve the manuscript substantially. We hope that the revised version is now suitable for publication in Communications Biology and remain open to any further suggestions.

Response to Reviewers' comments

We sincerely thank the reviewers for their thorough evaluation of our manuscript and for providing valuable, constructive feedback. We have carefully revised the manuscript accordingly and provide detailed responses to each comment below.

Reviewer #1 (Remarks to the Author):

The authors have addressed my comments well and the review is now much better structured.

As a final improvement to the section on oncogene signaling and Table 1, the authors should also consider including reported effects of up-regulating Cyclin D activity (through AMBRA1 depletion or Cyclin D1 overexpression, PMID: 33854232) and E2F activity (through E2F6 depletion, PMID: 32665547) on fork acceleration.

Response: Thank you for this suggestion. We have added the references you suggested into the main text and updated Table 1 to include the effects of Cyclin D and E2F activity on fork acceleration.

Reviewer #2 (Remarks to the Author):

The authors have answered my concerns and greatly improved the flow and the clarity of the manuscript. Thank you!

Response: We sincerely appreciate your positive feedback and are glad that the revisions improved the clarity and flow of the manuscript.